# A comprehensive *in silico* analysis of the deleterious nonsynonymous SNPs of human FOXP2 protein

**Mahmuda Akter**[1], **Sumaiya Farah Khan**[1], **Abu Ashfaqur Sajib**[2], **Fahmida Sultana Rima**[3]*

**1** Department of Genetic Engineering and Biotechnology, Jagannath University, Dhaka, Bangladesh,
**2** Department of Genetic Engineering and Biotechnology, University of Dhaka, Dhaka, Bangladesh,
**3** Department of Biochemistry and Biotechnology, University of Barishal, Barishal, Bangladesh

* fahmidarima7@gmail.com

**Data Availability Statement:** All relevant data are within the article and its Supporting Information files.

**Funding:** The authors received no specific funding for this work.

## Abstract

*FOXP2* encodes the forkhead transcription factor that plays a significant role in language development. Single nucleotide polymorphisms in *FOXP2* have been linked to speech- language disorder, autism, cancer and schizophrenia. So, scrutinizing the functional SNPs to better understand their association in disease is an uphill task. The purpose of the current study was to identify the missense SNPs which have detrimental structural and functional effects on the FOXP2 protein. Multiple computational tools were employed to investigate the deleterious role of non-synonymous SNPs. Five variants as Y531H, L558P, R536G and R553C were found to be associated with diseases and located at the forkhead domain of the FOXP2 protein. Molecular docking analysis of FOXP2 DNA binding domain with its most common target sequence 5'-CAAATT-3' predicted that R553C and L558P mutant variants destabilize protein structure by changing protein-DNA interface interactions and disruption of hydrogen bonds that may reduce the specificity and affinity of the binding. Further experimental investigations may need to verify whether this kind of structural and functional variations dysregulate protein activities and induce formation of disease.

## Introduction

*One of the members of the winged-helix (FOX) family of transcription factors is FOXP2 that plays a crucial role in language processing and cognition* [1]. During fetal development, the protein is expressed highly in the brain and several other parts of the body like the gut and lung [2]. The protein is characterized by a DNA-binding domain of forkhead box, large polyglutamine sequence, a conserved leucine zipper and zinc finger, acidic C-terminus and a dimerization domain [3]. The protein act as a transcription factor which is evolutionarily very conserved and may bind to near about 300–400 gene promoters in the human genome. Being a transcription element, FOXP2 can control a number of genes named *DISC1*, *CNTNAP2* and *SRPX2/µPAR*, that are likely to be considered as important players for speech and language development [4]. An experiment on mice suggested the role of FOXP2 in the process of

**Competing interests:** The authors have declared that no competing interests exist.

neuron formation, differentiation and migration [5]. The neural correlation of FOXP2 was shown in patients with verbal dyspraxia caused by disruption of the FOX domain [6].

All sorts of mutations have been described in *FOXP2* gene. For example, missense mutations [7], non-sense mutations [8], deletions [9,10] and indels [11] have been discovered in different cases and families [12]. Single nucleotide polymorphism in the enciphering area of the *FOXP2* gene leads to problems like speech-language disorder 1 (SPCH1) and developmental apraxia of speech (DAS) with impaired language communication and comprehension [13]. Polymorphisms of the gene, located at the chromosome 7q31, have also been related with frontotemporal lobar degeneration [14], schizophrenia [15], autism disorder [16] and development and progression of various types of cancer such as ovarian [17], breast [18] and prostate [19] cancer.

Single nucleotide polymorphisms (SNPs) in the human genome are most available genetic variations that are single base pair changes found in every 200–300 base pairs and act as genetic markers [20]. Approximately 0.5 million SNPs, that reside in the open reading frames of the human genome [21] have drawn much interest because they produce a large number of amino acid variations that lead to functionally diverse protein variants many of which eventually lead to diseases [22]. Several previous investigations have shown that more than fifty percent of the variations associated with hereditary genetic disorders are caused by substitution of amino acids known as non-synonymous SNPs (nsSNPs) [23]. These functional variations can exert deleterious or neutral effects on protein structure or function [24]. Detrimental effects of nsSNPs might cause destabilizing protein structure, alteration of gene regulation, changing ligand-binding site [25], changing protein hydrophobicity, geometry, charge [26], modify dynamics, stability, protein-protein interactions and alter translation, resulting to threatening the cellular structural integrity [27]. It has also been reported that the involvement of multiple nsSNPs stimulate the possibility of infections, autoimmune diseases and the expansion of inflammatory disorders [20].

Though over the past couple of years, various computational methods and techniques have been applied for sorting and detecting the effects of disease-associated SNPs in other subfamilies of the target protein, limited investigations have been performed on FOXP2 protein. Considering the significant role of the protein, the present research work has been designed to identify the deleterious nsSNPs and assess their pathogenic effects on the protein using various *in silico* algorithms. Since FOXP2 regulates the transcription of various target genes; molecular docking was performed to understand the effects of the variants on target DNA binding that may lead to altered gene regulation.

## Materials and methods

### Retrieval of missense SNPs

The list of all missense SNPs of *FOXP2* gene, relevant information (reference SNP ID, position, changed amino acid residue and protein accession number) of the gene and corresponding protein were retrieved from NCBI dbSNP (https://www.ncbi.nlm.nih.gov/projects/SNP/) and UniProt (https://www.uniprot.org/) databases [28]. Only the nsSNPs were considered for further exploration.

### Identifying the most detrimental nsSNPs

The substitution of amino acid at a particular position often greatly affects the function of the protein. Five various computational tools-PROVEAN [29], PolyPhen-2 [30], SNPnexus [31], SNAP2 [32] and PON-P2 were employed to anticipate the effect of nsSNPs on FOXP2 function [33]. Based on position-specific independent count (PSIC) scores, PolyPhen-2 estimates

the impact of mutation on protein at both structural and functional levels. The difference between the scores clarifies the nsSNPs into benign, possibly damaging and probably damaging [34]. An open accessible web server PROVEAN considers the list of mutant variants and generates a score after homology searching. The threshold value ≥-2.5 labelled the SNPs into neutral or deleterious [29]. SNPnexus is comprised of with the combination of SIFT and Poly-Phen tools [31]. SIFT defines the variants as deleterious when the tolerance index score is ≤0.05. Accessible properties of the wild and mutant protein were compared by the neural network based *in silico* tool SNAP2. The tool generates a heat map and differentiates the variants as effect or neutral providing a confidence score [32]. PON-P2, a machine learning-based server, computes random forest probability score to categorize the harmful variants into unknown, neutral or pathogenic [33].

## Prediction of disease related SNPs

Both SNPs&GO (DOI: 10.1002/humu.21047) and PhD-SNP (DOI: 10.1093/bioinformatics/btl423) online tools were applied to predict whether SNPs are disease associated or not. Support vector machine-based method SNPs&GO predicts the relationship of SNPs with disease at 81% accuracy. A probability score ≥0.5 points out that SNPs are related with clinical complications [35]. An online tool, PhD-SNP was also used to assess the connection of SNPs with diseases and classify them into neutral or disease associated on a 0–9 grade reliability index score [36].

## Alteration of structural and functional characteristics

MutPred 2 (http://mutpred.mutdb.org/) was used to predict the impact of disease associated SNPs based on the changes in 14 different biophysical properties. The tool evaluated the possibilities of addition or loss of some of the features providing a p-values. The p-values <0.05 and <0.01 were denote as significant and very significant output respectively [37].

## Protein stability prediction

A pair of software was used to check whether alterations of amino acid affect the protein stability. I-Mutant 2.0 (http://folding.biofold.org/i-mutant/i-mutant2.0.html) utilizes support vector machine to analyze any change in the stable state of the protein. The tool was derived from ProTherm which is a broad dataset of experimental data on protein mutations [38]. Conditions for all selected inputs were fixed at temperature $25^0$ C and pH 7.0. The outcomes provide a free energy change (DDG) and unfolding free energy value of the mutated and wild proteins [36]. MUpro (https://www.ics.uci.edu/~baldig/mutation.html) identifies the impact of single-site mutations in protein stability. A confidence score <0 dictates the reduction in stability of the protein while the score >0 interprets the opposite effect [39].

## Evolutionary conservation analysis

ConSurf (https://consurf.tau.ac.il/) is an effective tool that addresses the evolutionary pattern of conservation at each amino acid site [40]. By using the Bayesian calculation method, the server analyzes close sequence homologues and estimates the phylogenetic relationship [40]. For each residue of the candidate protein, the degree of conservation is estimated on a scale of 1 to 9 and classifies these as variable (1–4), intermediate (5–6) and conserved (7–9) [41].

## Prediction of post translational modifications sites (PTMs)

Diversified cellular functions like signaling cascade and protein-protein interactions are regulated by post-translational modifications (PTMs) of protein [42]. Since mutated residues can

induce allosteric or orthosteric effects that lead to shifts in energy conformations and stabilization, a deep understanding of the structure encompassing PTM sites helps to clarify the impact of PTMs on protein folding [43]. Phosphorylation, methylation, ubiquitination, acetylation, N-linked glycosylation and palmitoylation are some of the remarkable PTM that play crucial role in the study of diseases [44]. ModPred (http://www.modpred.org/), a sequence-based predictor, was applied to predict the overall influence of highly risky nsSNPs on the PTM of the target protein. This database evaluates the tendency of a specific amino acid to be modified [43].

## Molecular docking between FOXP2 variants and the consensus DNA motif

RaptorX, a deep-learning based structure prediction tool was applied to build the three dimensional structures of the wild type as well as the variants (L558P, R536G, R553C and Y531H) of FOXP2 DNA binding domain (DBD) [45]. Quality of the structures were assessed using ProSA-web (PMID: 17517781) and Ramachandran plot through PDBsum (PMID: 9433130) at the EMBL-EBI site. 3D structure (in Protein Data Bank or pdb format) of the most common target site of FOXP2 (5'-CAAATT-3') in B-form was generated using the web-based DNA Sequence to Structure conversion tool [46] at the Supercomputing Facility for Bioinformatics and Computational Biology, IIT, Delhi. Binding of FOXP2 DBD 3D models to CAAATT was predicted with HDock (PMID: 28521030). DNA docked structures of FOXP2 were further analyzed using Discovery Studio (v20.1.0.19295) tool (Discovery Studio Visualizer v20.1.0.19295, San Diego, USA) to identify the interacting residues of FOXP2 with its consensus DNA motif.

## Results and discussion

### Retrieval of nsSNPs dataset

The dbSNP database of NCBI provides a total of 144643 SNPs data for the FOXP2 protein. Out of 144643 SNPs, 141823 were reported to be present in intron region (98.05%), 393 were missense SNPs (0.27%), 240 were synonymous (0.17%) and 2353 were non-coding transcript variant (1.63%). Since the non-synonymous SNPs often alter the encoded amino acid, the present study only considered these SNPs for further analysis. The detail information about these SNPs was given in S1 Table. The proportions of each of the SNPs category are represented in graphical form (Fig 1).

### Identification of high-risk SNPs

Five computational algorithms like PolyPhen2, PROVEAN, SNPnexus, SNAP-2, PON-P2 were applied in the present study to identify detrimental nsSNPs in human FOXP2 protein. The identification outcomes with these tools are presented in Fig 2. Among the tools, Poly-Phen-2 anticipated 257 missense SNPs as probably or possibly damaging while PROVEAN determined 136 nsSNPs have deleterious effect on the proteins function. From the SIFT data of SNPnexus, 123 nsSNPs were found to be damaging. SNAP2 identified nonneutral 190 nsSNPs that might cause change in protein's activity while PON-P2 estimated 109 nsSNPs as pathogenic. The list of these SNPs is provided in S2 Table. Out of the 393 missense mutations, potent 43 nsSNPs that are found to be highly risky in all of the prediction tools were prioritized for further analysis and listed in Table 1.

### Identification of disease associated deleterious SNPs

The deleterious polymorphisms were then subjected to SNPs&GO and PhD-SNP servers to predict the variants that are potentially associated with diseases or not. Out of the 43 high

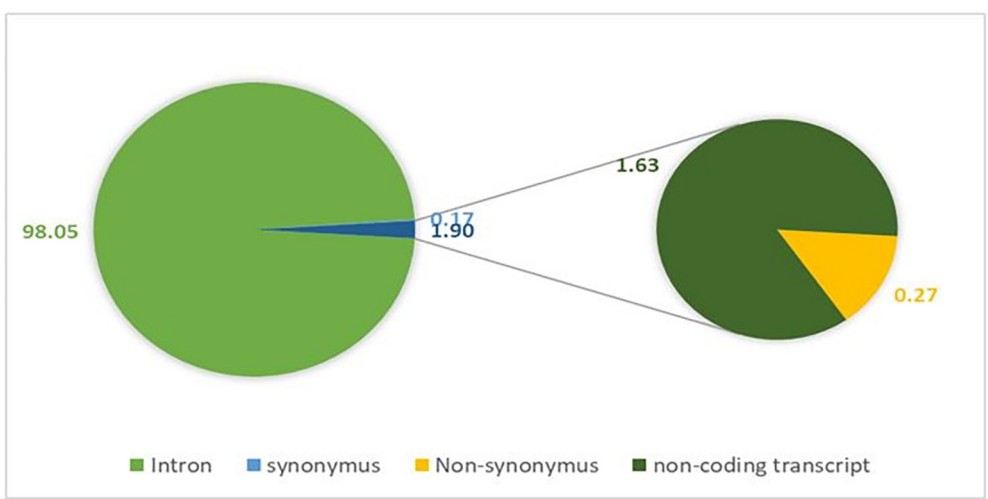

**Fig 1. Percentages of various kinds of SNPs in human FOXP2 protein.** Intronic SNPs: 98.05%; nsSNPs: 0.28%; synonymous SNPs: 0.17%; non-coding transcript: 1.63%.

confidence nsSNPs, SNPs&GO and PhD-SNP server designated 11 and 14 nsSNPs to have potential role in diseases. Among these, 5 nsSNPs were common in both tools and selected for further investigations (Table 2).

## Functional and structural modification prediction

The selected disease related nsSNPs were analyzed by MutPred 2 server to understand their molecular mechanism of pathogenicity (Table 3). The predicted modifications of structural and functional properties include alteration of disordered interface, transmembrane protein,

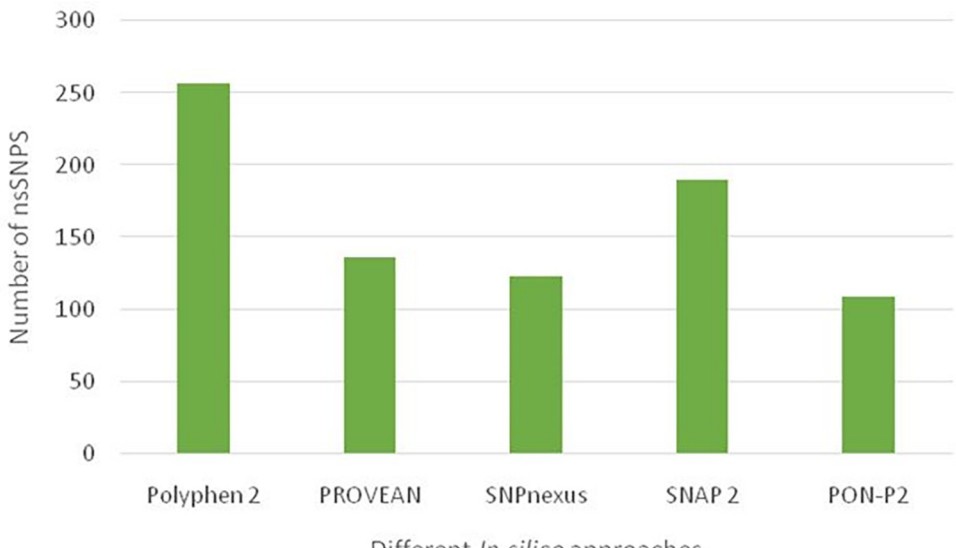

**Fig 2. Number of predicted damaging nsSNPs in human FOXP2.** The highest number of damaging nsSNPs were screened by Polyphen 2 and PON-P2 predicted the lowest number.

**Table 1. High-risk non-synonymous SNPs predicted by five computational tools.**

| rsID | Position | Provean | SIFT (SNPnexus) | Polyphen-2 | SNAP2 | PON P2 |
|------|----------|---------|-----------------|------------|-------|--------|
| rs121908377 | R553H | Deleterious | Deleterious | Probably damaging | Effect | Pathogenic |
| rs797045587 | M94I | Deleterious | Deleterious | Probably damaging | Effect | Pathogenic |
| rs879253772 | Y531H | Deleterious | Deleterious | Probably damaging | Effect | Pathogenic |
| rs112732214 | L558P | Deleterious | Deleterious | Probably damaging | Effect | Pathogenic |
| rs199776572 | T451M | Deleterious | Deleterious | Probably damaging | Effect | Pathogenic |
| rs377420314 | Y604C | Deleterious | Deleterious | Probably damaging | Effect | Pathogenic |
| rs566961630 | R553C | Deleterious | Deleterious | Probably damaging | Effect | Pathogenic |
| rs745342916 | D375G | Deleterious | Deleterious | Probably damaging | Effect | Pathogenic |
| rs751931499 | V690G | Deleterious | Deleterious | Probably damaging | Effect | pathogenic |
| rs755297474 | E334V | Deleterious | Deleterious | Probably damaging | Effect | Pathogenic |
| rs758427088 | D644H | Deleterious | Deleterious | Possibly damaging | Effect | Pathogenic |
| rs758513311 | S75F | Deleterious | Deleterious | Probably damaging | Effect | Pathogenic |
| rs762564041 | R536G | Deleterious | Deleterious | Probably damaging | Effect | Pathogenic |
| rs765157455 | G319E | Deleterious | Deleterious | Probably damaging | Effect | Pathogenic |
| rs765157455 | Q117H | Deleterious | Deleterious | Probably damaging | Effect | Pathogenic |
| rs772694863 | W270R | Deleterious | Deleterious | Probably damaging | Effect | Pathogenic |
| rs779754644 | S305F | Deleterious | Deleterious | Possibly damaging | Effect | Pathogenic |
| rs779921362 | E676K | Deleterious | Deleterious | Possibly damaging | Effect | Pathogenic |
| rs889341368 | G473R | Deleterious | Deleterious | Probably damaging | Effect | Pathogenic |
| rs948249504 | S255W | Deleterious | Deleterious | Probably damaging | Effect | Pathogenic |
| rs1158865993 | P697L | Deleterious | Deleterious | Probably damaging | Effect | Pathogenic |
| rs1175210435 | P115S | Deleterious | Deleterious | Probably damaging | Effect | Pathogenic |
| rs1183578823 | R376L | Deleterious | Deleterious | Possibly damaging | Effect | Pathogenic |
| rs1184039801 | W270C | Deleterious | Deleterious | Probably damaging | Effect | Pathogenic |
| rs1191637371 | N567I | Deleterious | Deleterious | Probably damaging | Effect | Pathogenic |
| rs1219589831 | N694Y | Deleterious | Deleterious | Probably damaging | Effect | Pathogenic |
| rs1224336230 | V563A | Deleterious | Deleterious | Probably damaging | Effect | Pathogenic |
| rs1250616703 | L269F | Deleterious | Deleterious | Probably damaging | Effect | Pathogenic |
| rs1276471970 | N424K | Deleterious | Deleterious | Probably damaging | Effect | Pathogenic |
| rs1290398957 | H340Y | Deleterious | Deleterious | Possibly damaging | Effect | Pathogenic |
| rs1295112601 | L236F | Deleterious | Deleterious | Possibly damaging | Effect | Pathogenic |
| rs1302127838 | M406K | Deleterious | Deleterious | Possibly damaging | Effect | Pathogenic |
| rs1330529378 | P677S | Deleterious | Deleterious | Probably damaging | Effect | Pathogenic |
| rs1343377230 | F507L | Deleterious | Deleterious | Probably damaging | Effect | Pathogenic |
| rs1362466494 | L291P | Deleterious | Deleterious | Probably damaging | Effect | Pathogenic |
| rs1375575897 | Y604N | Deleterious | Deleterious | Probably damaging | Effect | Pathogenic |
| rs1383948441 | S315F | Deleterious | Deleterious | Probably damaging | Effect | Pathogenic |
| rs1394757420 | V112F | Deleterious | Deleterious | Probably damaging | Effect | Pathogenic |
| rs1428334171 | P486H | Deleterious | Deleterious | Probably damaging | Effect | Pathogenic |
| rs1436939063 | E334K | Deleterious | Deleterious | Possibly damaging | Effect | Pathogenic |
| rs1445779721 | K365M | Deleterious | Deleterious | Probably damaging | Effect | Pathogenic |
| rs1447805795 | E334K | Deleterious | Deleterious | Possibly damaging | Effect | Pathogenic |
| rs1459605752 | H651R | Deleterious | Deleterious | Possibly damaging | Effect | Pathogenic |

metal binding, coiled coil, stability, gain of loop, B factor, intrinsic disorder, amidation, loss of helix as well as loss and gain of glycosylation. L291P variant significantly gain GPI-anchor amidation and N-linked glycosylation. L558P was predicted to induce loss of helix and N-linked

**Table 2. Disease related non-synonymous SNPs identified by SNPs&GO and PhD-SNP.**

| rsID | Variant | Mutation | SNP&GO | PhD-SNP |
|---|---|---|---|---|
| rs879253772 | T/C | Y531H | Disease | Disease |
| rs112732214 | T/C | L558P | Disease | Disease |
| rs566961630 | C/T | R553C | Disease | Disease |
| rs762564041 | C/T | R536G | Disease | Disease |
| rs1362466494 | T/C | L291P | Disease | Disease |

RI = Reliability Index, P = Probability.

glycosylation with p value 0.04 and 0.05 respectively. R536G variants also predicted to have loss of N-linked glycosylation.

## Effect on protein stability

Both I-Mutant 2.0 and MUpro servers were used to interpret whether the protein will be in stable or denatured form due to point mutation. The result expressed either in the free energy change value ($\Delta\Delta G$) or in the sign of DDG where positive and negative DDG value indicates increased or decreased stability, respectively. All of the variants represented in Table 4 have negative DDG value indicating that they reduce the stability of the variants.

**Table 3. Impacts of nsSNPs on structural & functional properties of FOXP2.**

| Mutation | Probability of deleterious mutation | Structural & functional properties |
|---|---|---|
| Y531H | 0.897 | Altered coil coil (P = 0.53) |
| | | Altered ordered interface (P = 0.37) |
| | | Altered disordered interface (P = 0.31) |
| | | Altered transmembrane protein (P = 0.25) |
| L291P | 0.573 | Gain of intrinsic disorder (P = 0.40) |
| | | Gain of loop (P = 0.27) |
| | | Gain of B factor (P = 0.1797) |
| | | Gain of GPI-anchor amidation at N-295 **(P = 0.05)** |
| | | Gain of N linked glycosylation at N294 **(P = 0.02)** |
| L558P | 0.923 | Loss of helix (P = 0.04) |
| | | Altered disordered interface (P = 0.24) |
| | | Altered transmembrane protein (P = 0.20) |
| | | Altered Stability (P = 0.12) |
| | | Loss of N linked glycosylation at N555 **(P = 0.05)** |
| R553C | 0.9 | Altered Metal binding (P = 0.39) |
| | | Altered Disordered interface (P = 0.38) |
| | | Loss of Helix (P = 0.29) |
| | | Altered Ordered interface (P = 0.27) |
| | | Altered Coiled coil (P = 0.25) |
| R536G | 0.914 | Altered Transmembrane protein (P = 0.18) |
| | | Loss of N-linked glycosylation at N555 **(P = 0.04)** |
| | | Altered Coiled coil (P = 0.48) |
| | | Altered Disordered interface (P = 0.42) |
| | | Altered Transmembrane protein (P = 0.31) |
| | | Altered Ordered interface (P = 0.30) |
| | | Loss of Helix (P = 0.29) |

**Table 4. Impact of single nucleotide polymorphisms on the stability of protein.**

| dbSNP | Amino acid changes | Mu-pro | | I-Mutant | |
|---|---|---|---|---|---|
| | | Prediction | DDG value | Prediction | DDG value |
| rs879253772 | Y531H | Decrease | -1.244777 | Decrease | -1.34 |
| rs112732214 | L558P | Decrease | -2.0758766 | Decrease | -1.77 |
| rs566961630 | R553C | Decrease | -0.71979678 | Decrease | -0.57 |
| rs762564041 | R536G | Decrease | -1.2125418 | Decrease | -1.63 |
| rs1362466494 | L291P | Decrease | -1.4678293 | Decrease | -1.24 |

DDG = Free energy change value, RI = Reliability index.

### Prediction of post translational modifications

Out of 5 nsSNPs, the Modpred server reported only 2 nsSNPs (Y531H and R536G) to be present at PTM sites and 1 variant (R553C) at amidation sites. Table 4 depicts the post translational modification (PTM) of altered amino acids.

### Sequence conservation study of nsSNPs

The variants Y531H, L558P, R536G and L291P were reported by consurf to be highly conserved as structural and buried and R553C was predicted to be conserved being functional and exposed. The server estimated the confidence level by giving a conservation score (1–9) for the conservation of sequence. The functional and structural consequences of 5 high-risk nsSNPs with their phylogenetic conservation scores are given in S1 Fig and Table 5.

### Structural validation of wild and mutant models

The four mutants Y531H, L558P, R553C, R536G are present in the winged-helix DNA binding domain (DBD) of the FOXP2 protein. So, the native DBD structure of the protein and its four detrimental, disease associated variants were using RaptorX. Model quality validation output file of Ramachandran plot and ProSA Z- score were described in Table 6. The Ramachandran plot depicted that 96.1% residues were in most favored region whereas 0% in disallowed region among the 82 (503–584) residues in native DBD of the FOXP2 protein. Over 94% residues also occupy most favorable region in the predicted DBD structure of the 4 variants. ProSA Z- score and Ramachandran plot indicate the overall good quality of the 3D modelling.

**Docking analysis.** The docking analysis of FOXP2 DBD with its most common target sequence reported that the mutant variants bind to the DNA sequence in a slightly diverged orientation than wild type variants of DBD. Upon binding, the native protein shows interaction with Arg 504, Tyr 509, His 554, Leu 558, Pro 505, Trp 548 and Ala 551residues. Positively

**Table 5. Conservation profile of nsSNPs in FOXP2 protein and their post translational modification.**

| rsID | Residue & position | Conservation score | Prediction | PTMs |
|---|---|---|---|---|
| rs879253772 | Y531H | 9 | Highly conserved and buried (s) | Proteolytic cleavage |
| rs112732214 | L558P | 9 | Highly conserved and buried (s) | - |
| rs566961630 | R553C | 9 | Highly conserved and exposed (f) | Amidation |
| rs762564041 | R536G | 9 | Highly conserved and buried (s) | Proteolytic cleavage |
| rs1362466494 | L291P | 9 | Highly conserved and buried (s) | |

PTMs = Post translational modification.

**Table 6. The results of three-dimensional model validation.**

| PDBSum | Most favoured region (%) | Additional allowed region (%) | Generously allowed region (%) | Disallowed region (%) | ProSA web (Z-score) |
|---|---|---|---|---|---|
| FOXP2 DBD (WT) | 96.1 | 2.6 | 1.3 | 0 | -5.81 |
| FOXP2 DBD (L558P) | 94.7 | 5.3 | 0 | 0 | -6.5 |
| FOXP2 DBD (R536G) | 97.4 | 1.3 | 1.3 | 0 | -6.22 |
| FOXP2 DBD (R553C) | 96.1 | 2.6 | 0 | 1.3 | -6.2 |
| FOXP2 DBD (Y531H) | 98.7 | 0 | 1.3 | 0 | -6.18 |

charged Arg 504 shows electrostatic interaction with Thymine10′ and Tyr 509 forms hydrogen bond with Thymine5. All of the variants form this type of bond with Arg 504 and Tyr 509 on average 0.05 or marginal deviation of distance respectively. Y531H missense variant interacts with the same amino acid residues except Ala 551. In native protein, Ala 551 form pi-alkyl type hydrophobic bond with Adenine8′ but Adenine8′ forms carbon hydrogen bond to Thr 547 in Y531H variants. Thr 547 makes contact with Adenine8′ and due to substitution of Leu to Pro, Thymine6 binds with Pro 558 and His 559 in L558P mutants. Similar pattern of interactions was also observed in R536G variants. Due to alteration of arginine to cysteine, Phe forms Pi-Amino bond with Thymine10′ at position 553. The details of the interactions have been described in Fig 3 and S3 Table.

The winged helix DNA binding domain of FOXP2 is crucial for analysis as majority of the disease-causing mutations are found in that region [47]. The mutation of arginine to histidine at 553th position in human FOXP2 has been associated with severe congenital speech impairement disorder which plays a vital role in the FOXP2-DNA interaction [48]. Most of the interactions between deoxyribonucleic acid and transcription factors are between the negatively charged phosphodiester bond of the DNA and positively charged amino acid residues of the protein [49]. Banerjee-Basu and Baxervanis in 2004 observed that the R553H mutation causes a net reduction of the electrostatic potential in the DNA binding surface of the protein. Due to the disturbance to the electrostatics of the protein the interactions between the backbone of the DNA and protein might be hampered. Several studies also suggest that electrostatic interactions also dominate the protein-DNA binding specificity [50]. In case of R553C mutation, a new electrostatic bond is formed between Phe 541 and Thy 10′ which may change the electrostatic potential that likely led to imbalance in the organization of protein -DNA interface.

From various mutational investigations it was known that His554 plays a significant role in hydrogen bond formation between FOXP2 FHD and DNA [51]. This interaction was also conserved in other FOX DBDs like FOXO1, FOXK1 and FoxA1 [52]. The substitution of leucine to proline at 558th position missed the interaction with His554, which may lead to the reduction of specificity of the protein with its cognate sequence. Since hydrogen bond stabilizes the protein-DNA interaction [53], the missing of most conserved H-bond might destabilize the interaction. The more interactions between the DNA and protein may predict the higher affinity of binding. Upon binding, the native protein shows interaction with Arg 504, Tyr 509, His 554, Leu 558, Pro 505, Trp 548, Ala 551 residues. L558P mutation reduces the interactions with other residues compared to native protein which may lead to lower affinity of protein-DNA binding.

## Conclusion

With the combination of various *in silico* tools, the study determined the deleterious nsSNPs in FOXP2 protein. The outcomes predicted R553C and L558P mutants as the most damaging

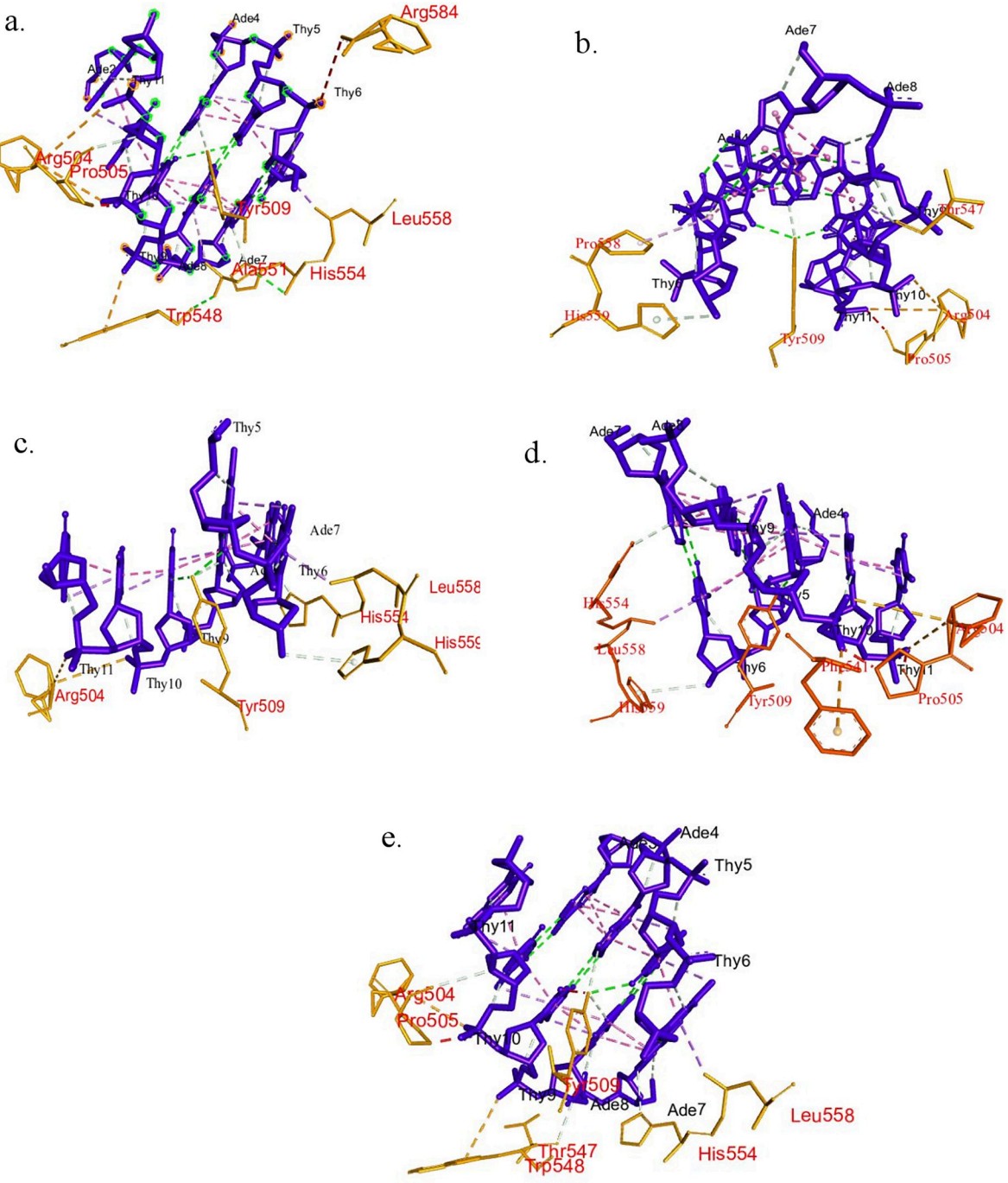

**Fig 3. Molecular docking analysis of native and mutant DBD domain of FOXP2 with novel sequence.** Interactions of DNA binding domain with (a) Native FOXP2 (b) Mutant L558P (c) R536G (d) R553C (e) Y531H.

nsSNPs that may alter protein-DNA interface interactions, reduce the specificity and affinity of the binding. This kind of structural and functional alterations is likely to interrupt regulation of protein and initiate formation of disease. In-depth experimental verification is needed to determine the role of mutant variants more precisely.

## Supporting information

**S1 Fig. Prediction of ConSurf about the conservation status of amino acids in human FOXP2 protein.**
(TIFF)

**S1 Table. The list of reference sequence ID, alteration of allele, protein accession no, position of amino acid and residual change of all 393 non-synonymous SNPs.**
(DOCX)

**S2 Table. The results of five *in silico* tools that are used to analyze all of the 393 non-synonymous SNPs.**
(DOCX)

**S3 Table. Interactions of amino acid residues with native and mutant FoxP2 protein.**
(DOCX)

## Author Contributions

**Conceptualization:** Sumaiya Farah Khan, Fahmida Sultana Rima.

**Data curation:** Mahmuda Akter, Sumaiya Farah Khan, Abu Ashfaqur Sajib, Fahmida Sultana Rima.

**Formal analysis:** Mahmuda Akter, Abu Ashfaqur Sajib.

**Methodology:** Sumaiya Farah Khan, Fahmida Sultana Rima.

**Supervision:** Sumaiya Farah Khan, Abu Ashfaqur Sajib.

**Validation:** Fahmida Sultana Rima.

**Writing – original draft:** Mahmuda Akter.

**Writing – review & editing:** Sumaiya Farah Khan, Abu Ashfaqur Sajib.

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
