## [Decision Letter · Decision Letter 0]

25 Jul 2022

A comprehensive in silico analysis of the deleterious nonsynonymous SNPs of human FOXP2 protein

PONE-D-22-09322

Dear Dr. Rima,

We’re pleased to inform you that your manuscript has been judged scientifically suitable for publication and will be formally accepted for publication once it meets all outstanding technical requirements.

Kind regards,

Michael Massiah

Academic Editor

PLOS ONE

Additional Editor Comments (optional):

PS. I would also like to apologize for the tardiness of the review process. It has been a challenge finding reviewers.

Reviewers' comments:

Reviewer's Responses to Questions

**Comments to the Author**

1. Is the manuscript technically sound, and do the data support the conclusions?

Reviewer #1: Partly

2. Has the statistical analysis been performed appropriately and rigorously? 

Reviewer #1: N/A

3. Have the authors made all data underlying the findings in their manuscript fully available?

Reviewer #1: Yes

4. Is the manuscript presented in an intelligible fashion and written in standard English?

Reviewer #1: Yes

5. Review Comments to the Author

Reviewer #1: Title: A comprehensive in silico analysis of the deleterious nonsynonymous SNPs of human

FOXP2 protein

Fahmida Sultana Rim et al determined the deleterious nsSNPs in FOXP2 protein. Overall manuscript is nicely written and accepted for the publication.

6. PLOS authors have the option to publish the peer review history of their article (what does this mean?). If published, this will include your full peer review and any attached files.

Reviewer #1: **Yes: **Dr.Kinnari N Mistry, Associate Professor, Ashok & Rita Patel Institute of Integrated Study & Research In Biotechnology And Allied Sciences, Affiliated to CVM University, New Vallabh Vidyanagar, Gujarat, India

---

## [Editor Report · Acceptance letter]

29 Jul 2022

PONE-D-22-09322 

A comprehensive *in silico* analysis of the deleterious nonsynonymous SNPs of human FOXP2 protein 

Dear Dr. Rima:

I'm pleased to inform you that your manuscript has been deemed suitable for publication in PLOS ONE. Congratulations! Your manuscript is now with our production department. 

Kind regards, 

on behalf of

Dr. Michael Massiah 

Academic Editor

PLOS ONE